# Optimization of Carsharing Fleet Placement in Round-Trip Carsharing Service

**Boonyarit Changaival** [1,†]**, Kittichai Lavangnananda** [2,*,†]**, Grégoire Danoy** [1,†]**, Dzmitry Kliazovich** [3,†]**, Frédéric Guinand** [4,†]**, Matthias Brust** [5,†]**, Jedrzej Musial** [6,†] **and Pascal Bouvry** [1,†]

1  Interdisciplinary Centre for Security, Reliability and Trust (SnT)-Faculty of Science, Technology and Medicine/Computer Science and Communications(FSTM/CSC), University of Luxembourg, Esch-sur-Alzette, 4364 Luxembourg, Luxembourg; b.c.boonyarit@gmail.com (B.C.); gregoire.danoy@uni.lu (G.D.); pascal.bouvry@uni.lu (P.B.)
2  School of Information Technology (SIT), King Mongkut's University of Technology Thonburi, Bangkok 10140, Thailand
3  ExaMotive S.A., Esch-sur-Alzette, 4263 Luxembourg, Luxembourg; kliazovich@ieee.org
4  Laboratoire d'Informatique, du Traitement de l'Information et des Systéme (LITIS), Faculty of Science and Technology, Normandie University, 76600 Le Harve, France; frederic.guinand@univ-lehavre.fr
5  Interdisciplinary Centre for Security, Reliability and Trust (SnT),  University of Luxembourg, Esch-sur-Alzette, 4364 Luxembourg, Luxembourg; matthias.brust@uni.lu
6  Faculty of Computing and Telecommunications, Poznan University of Technology, 60-965 Poznan, Poland; jedrzej.musial@cs.put.poznan.pl
*  Correspondence: kitt@sit.kmutt.ac.th
†  These authors contributed equally to this work.

**Abstract:** In a round-trip carsharing system, stations must be located in such a way that allow for maximum user coverage with the least walking distance as well as offer certain degrees of flexibility for returning. Therefore, a balance must be stricken between these factors. Providing a satisfactory system can be translated into an optimization problem and belongs to an NP-hard class. In this article, a novel optimization model for the round-trip carsharing fleet placement problem, called Fleet Placement Problem (FPP), is proposed. The optimization in this work is multiobjective and its NP-hard nature is proven. Three different optimization algorithms: PolySCIP (exact method), heuristics, and NSGA-II (metaheuristic) are investigated. This work adopts three real instances for the study, instead of their abstracts where they are most commonly used. They are two instance:, in the city of Luxembourg (smaller and larger) and a much larger instance in the city of Munich. Results from each algorithm are validated and compared with solution from human experts. Superiority of the proposed FPP model over the traditional methods is also demonstrated.

**Keywords:** carsharing system; fleet placement; metaheuristic algorithm; multiobjective optimization; NP-hard problem; NSGA-II; optimization; PolySCIP

## 1. Introduction

It is undeniable that efficient management of transportation has become one of the major problems in cities across the globe due to its impact on the environment and quality of life. Carsharing is one of many means of transportation nowadays and has received positive support from communities and governments. Its success can be seen in several countries, such as Germany, which has the biggest carsharing market in Europe with over 2 million registered users, 170 service providers, and over 16,000 vehicles available in 740 cities [1,2]. Coupling with the increasing awareness in environmental problems, the concept of green mobility is also promoted through the electrical carsharing service [3]. For instance, it has been highlighted that cars are used for transportation more than trains and planes in Germany and that carsharing positions itself is an intermediate mean to fill the gap between public transport and personal cars [4]. Another success was reported in

the United Kingdom where the government provided support to extend the user base to 600,000 individuals by 2020 to reduce traffic and parking problems [2].

The carsharing model can be divided into free-floating and station-based [2]. Free-floating carsharing offers the highest degree of flexibility to the users. They pick up the nearby vehicle to start a trip and drop it off anywhere in the city to end the trip. However, flexibility comes with a high operational cost for the company, which needs to maintain a high density of vehicles even in low-demand areas of the city in order to cope with low levels of utilization. Additionally, vehicles that end up in low demand areas need to be reallocated. An example of well-known free-floating carsharing companies are SHARE NOW (which is the merge of Car2Go and DriveNow) [1].

In station-based carsharing, fleet vehicles are stationed in densely populated areas of the city and need to be returned to one of these locations after completing the trip (one-way) or to the same pick up area (round-trip). As a result, station-based services are less flexible, with an advantage of easier implementation and management. Station-based services require fewer cars, as carsharing operators can place vehicles in densely populated and high-demand zones of the city only, and no fleet relocation is typically required. An example for easier implementation and management is taken from bike-sharing. The bicycles need to be relocated everyday to maintain the service function, which is relatively easy because of their sizes. However, car relocation is more difficult and more expensive, which is an even bigger burden to carsharing operators. In addition, with the trend of electronic cars, a charging station is easier to implement in the station-based services due to having fewer stations to implement than the free-float services.

Therefore, a satisfactory solution associated with carsharing is multifarious. The work in this article is concerned with maximum user coverage and ease of access to the service (i.e., shortest distance to a station and flexibility in returning). It comprises the following:

1. Developing a concept of station and their locations that maximize user coverage while giving a certain degree of flexibility when returning a car;
2. Maintaining the right balance between user coverage and ease of access to the service;
3. Considering or designing a suitable metric which can be used to determine the ease of access for users at a global scale.

In this work, we propose a new method attempting to optimize the fleet placement in the station-based round-trip, will will be the first to tackle fleet allocation in round-trip carsharing. The model of this problem is called fleet placement problem (FPP). Fleet placement is really tedious and is usually performed manually by experts and hence is prone to errors due to lack of precision. The proposed methodology aims to maximize customer coverage, while minimizing the maximum walking distance between customers and the nearest vehicle. These two objectives are in conflict, thus, resulting in a bi-objective problem. Unlike previous solutions, the proposed model incorporates highly detailed street-level map data containing footprints of the buildings. The contributions proposed in this work are: mathematical formulation of the novel fleet placement problem (FPP) and its NP-hardness proof, correlation analysis of the two problem objectives, and the comparison of results between the manual placement and state-of-the-art heuristic and metaheuristic algorithms.

The remainder of this paper is organized as follows. Related work on fleet location for carsharing services and similar location problems are presented in Section 2. The formulation of the FPP problem and the methodology to solve it are detailed in Section 3. In Sections 5 and 6 results from executions on real city instances are presented and discussed. Finally, conclusions and perspectives are provided in Section 7.

## 2. Related Work

In this section, the state-of-the-art on fleet placement and location problems, shared fleet placement, and the optimization methods used to address are analyzed.

## 2.1. Fleet Placement and Location Problems

In the following, the similarities of the fleet location problem with two classical optimization problems, i.e., the maximal covering location problem (MCLP) and the facility location problem (FLP) are discussed. These two problems and the proposed fleet placement problem (FPP) can be reduced to the set covering problem as shown in the proof in Section 3.

Church and ReVelle proposed the maximal covering location problem (MCLP) in 1974 [5] for facility and emergency siting. The objective is to maximize the partial coverage with a number of facilities, where each facility has a fixed coverage distance. MCLP is shown to be NP-hard, which means it becomes intractable and cannot be solved in an acceptable time by exact methods when the size of an instance is large [6]. MCLP has been applied in many real-world problems. Seargeant used MCLP as a base model for placing healthcare facilities based on the demographic data in the regions [7]. Schmid et al. formulate their ambulance siting problem as an MCLP with the integration of patients' data and traces of taxis in Vienna to estimate the traveling time to reach the patient [8]. Another example in telecommunication is from Ghaffarinasab et al. who proposed a bi-level version of the hub interdiction problem (also another variant of MCLP) [9]. MCLP was also extended to its multi-objective. Xiao et al. proposed a MCLP with two objectives which were facility cost and proximity minimization [10]. Kim et al. solved another bi-objective version of the MCLP where the aim was to maximize primary and backup coverage (overlapping coverage for reliability) [11]. Malekpoor et al. formulated the problem of electrification in a disaster relief camp as finding a set of locations to reduce the project cost and increase the share of systems between sites [12].

In the facility location problem (FLP), the objective is to find locations to place facilities to supply stores, while minimizing the maximum cost (p-center) or the average cost (p-median) [13]. In this problem, one constraint is to have all the stores covered while one store can be covered by only one facility [14]. One of the many interesting applications of the FLP is shown in [15] where they utilized the spatial information and studied the difference between the optimal facilities locations and the current ones. Another application is in siting rescue boat locations. It was modelled as a multi-objective problem which considers not only the response time to the incidents, but also the operating cost and working hours [16]. FLP was also used in telecommunication to find the location of GSM antennas as shown in [17,18].

These two problems are highly related to our fleet placement problem (FPP). The similarity between FPP and MCLP is that they both try to maximize the partial coverage, with a constraint of fixed coverage distance and fixed number of facilities. Meanwhile, the FLP objective is to minimize the maximum operating cost, which is well aligned with FPP, with a second objective, which is to minimize the maximum walking distance. Therefore, FPP can be seen as a combination of these two problems.

## 2.2. Shared Fleet Placement

Shared fleet placement can be formulated into MCLP or FLP (especially in round-trip carsharing service). However, there are other factors to be considered. In previous works on MCLP and FLP, they already have a list of preferred locations. These possible sites are evaluated by considering convenience factors such as parking cost, the proximity to essential facilities, and accessing time as presented in [19,20]. The solution was then a combination of selected sites to maximize user coverage. In fact, they are very similar to the facility location problem. Kumar and Bierlaire evaluated potential stations by the distance between the station and other facilities such as hospitals and train stations. They also had access to historical data to make a decision on where to place the station, which is not available in most cases [20]. Another popular approach is to locate the fleet by user demands [21–23]. Boyacı et al. proposed an optimization model to maximize the user coverage based on the demand and predicted destinations [21]. On the contrary, Lage et al. studied a method to identify the potential of city districts in a station-based one-way trip

scenario where the demands were estimated from the taxi trips and customer profiles in Sao Paulo, Brazil [22]. Lastly, Schwer and Timpf proposed an idea for locating the fleet in round-trip carsharing by combining both user demands and proximity to other mean of transportation and other facilities into a model and utilized the open source data available from the government [23].

There are also works which focus on electric carsharing fleets, reflecting the increasing awareness of environmental issues and benefiting from governmental support. The electric carsharing fleet is more complicated than its fossil-fuel counterpart considering the additional constraints for battery/electrical load management of the car. Çalik and Fortz proposed a model for a one-way electric carsharing service which considered previously mentioned factors [24]. Since charging is very important in electric carsharing service, Jiao et al. [25] formulated their model to consider a situation where the user changes the drop off station. The charging station location optimization was presented by Brandstätter et al. [26] where the authors based the location on the source and destination of trips in both simulation and Vienna. Another example was proposed by Yıldız et al. [27] which consider a more realistic case where demand was stochastic and capacitated charging stations. In addition, the shared fleet placement is also studied in the bikesharing community where station locations and bike stocking are highly important as well [28–32].

### 2.3. Existing Resolution Approaches

Several algorithms were proposed to solve the aforementioned problems, ranging from exact methods, to heuristic and metaheuristic algorithms. Exact methods guarantee optimality, however, once the size of an NP-hard problem is too large, such methods (e.g., branch-and-bound, exhaustive search) cannot find solutions in reasonable time. In contrast, heuristic algorithms (e.g., greedy algorithms) are problem specific methods that permit to obtain an approximate solution in reasonable time. Finally, metaheuristic algorithms (e.g., genetic algorithm or simulated annealing), are general purpose algorithms, which can lead to very satisfactory solutions. A true benefit is their acceptable execution time, which for middle to large size instances is several orders of magnitude smaller than for exact methods [33].

For generic location problems such as MCLP and FLP, exact methods are usually used [7,8,12,16,34–36]. It is important to note that the problem instances tackled in the reported articles were of limited size. In fact, Zarandi et al. [36] reported that IBM CPLEX [37] cannot handle a problem with a large size of input (e.g., a city). Hence, once the problem size is too large, heuristic and metaheuristic algorithms are usually employed. Church and ReVelle [5] first compared two variations of heuristic algorithms (which add one facility location one at a time) and a branch and bound algorithm. The next attempt in solving MCLP was using a Lagrangian heuristic algorithm, which is a combination of the Lagrangian Relaxation approach and a greedy method [38]. Heuristic algorithms are still being used nowadays as shown in [39] to solve the FLP problem. Lastly, several works reported on the efficiency of metaheuristics in solving FLP and MCLP. Tabu Search (TS), Simulated Annealing (SA), Variable Neighborhood Search (VNS), and Genetic Algorithms (GA) were also considered in solving MCLP [36,40–43]. Metaheuristics were not only used to solve single-objective versions of these problems but also multi-objective ones. Xiao et al. [10] employed a Multi-Objective Evolutionary Algorithm (MOEA) to solve the bi-objective MCLP, which focused on facility cost and proximity minimization using a specific encoding scheme and dedicated operators. Kim and Murray [11] solved the bi-objective reliability-focused MCLP where it aimed to maximize primary and backup locations coverage with a heuristic algorithm and a Multi-Objective Genetic Algorithm (MOGA). Karasakal and Silav [44] utilized the crowding distance function from the Non-dominated Sorting Genetic Algorithm II (NSGA-II) [45] in the Strength Pareto Evolutionary Algorithm II (SPEA2) [46] and reported that the new algorithm outperformed the original NSGA-II and SPEA-II. Ranjbartezenji et al. [47] proposed their modified version of NSGA-II and used it to solve bi-objective MCLP.

In shared fleet placement, the most common approach is to model it as a single-objective problem (weighted sum) and to rely on exact solvers such as CPLEX or MATLAB [20,21,25,48,49]. Several path-based heuristic algorithms were proposed [24,26]. In fact, due to the problem complexity, heuristic and metaheuristic algorithms have been attracting more attention recently [50]. Another approach is to determine the fleet locations through agent-based simulation [51].

To date, metaheuristic algorithms have not been applied to shared fleet placement problem. Previous works mainly either consider small-size instances of the problem (at most 800 potential locations by curation) and apply exact methods, or propose heuristic algorithms. The location curation is normally conducted by field experts and is essential to facilitate the applications of exact methods and proposed heuristic algorithms, but it can be very time consuming and prone to error. Therefore, we aim to eliminate the curation process (to be fully automated) which in turn, leaves decision makers to consider over 100,000 locations (in cities like Munich). General exact methods and previously proposed heuristic algorithms are too computationally expensive to apply. Therefore, metaheuristic algorithms are a suitable candidate to solve such big instances of fleet placement problem efficiently.

## 3. Optimization Model

The focus of this article is on the round-trip carsharing service, which received relatively less attention in the research community [52]. Hence, a novel approach for round-trip carsharing fleet placement is proposed. There are three benefits that the approach offers. The first benefit is that automation of the fleet placement process removes the need for traditional manual allocation. The second benefit is the higher placement precision with the inclusion of a Geographic Information System (GIS). Finally, the third benefit is in proposing an approachable fleet management problem to the research community that may also be beneficial to similar applications.

The proposed approach emphasizes the user coverage and ease of access with a constraint of being applicable to the real-world scenario by utilizing the two components mentioned below. These two goals are of real concern in practice and are often expressed by experts in this area.

1.  Utilizing graph theory to implement graph model representing a street network
2.  Multiobjective Optimization model with two objectives, maximizing user coverage and minimizing global walking distance between cars and users

In this section, the graph instance used in this article is first defined for the FPP since the graph instance is closely related with the problem definition. Second, the Fleet Placement Problem (FPP) model is formulated. Finally, FPP is proven that it is an NP-hard in a strong sense.

### 3.1. Graph Instance Definition

The street map can be modeled as an undirected weighted graph to represent users' ability to walk on the streets in both directions where they need to pick up or leave the shared vehicle (see Figure 1). The street-level graph is modeled as follows:

$$G = (V, E, P, W).$$

The set of nodes $V$ is composed of two subsets: $V = S \cup B$, where $S$ is the subset of street nodes (i.e., nodes on roads and streets), while subset $B$ contains buildings. Both $S$ and $B$ are of type node, hence they are naturally members of vertices in $G$.

Buildings contain users. A weight $p_i \in P$ associated with each node $b_i \in B$ corresponds to the estimation of the number of people living in this building. The weight for each node in $S$ is set to zero, making the assumption that target users are only located in buildings. This is similarly to the study taken by Daniels and Mulley [53]).

The set of edges *E* consists of two subsets: $E = R \cup L$, where *R* is a set of streets/roads, and *L* is a set of links connecting residential buildings to nearby streets.

Each edge $(u, v) \in R$ is valuated by a weight, $w_i \in W$, representing the walking distance from *u* to *v* where $u, v \in S$.

Each edge $(u, b) \in L$ where $u \in S$ and $b \in B$ is valuated by 0. In other words we consider the distance between the building and its nearest adjacent street is considered negligible. This process is called "snapping" and is common in every routing service where the starting point is first projected on a road before starting to build a route [54].

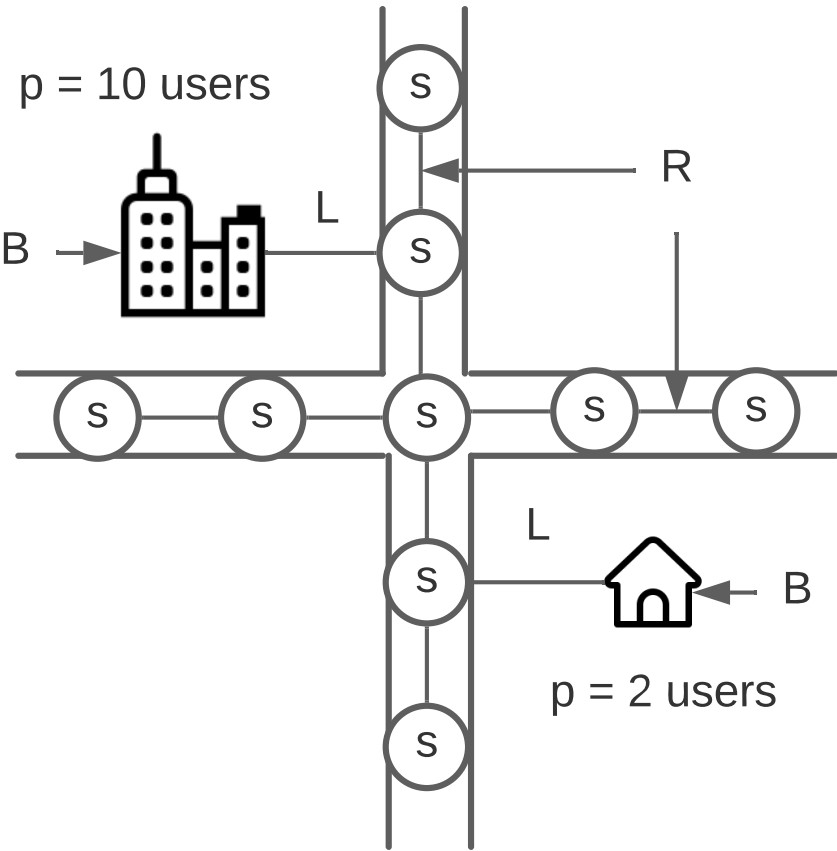

**Figure 1.** An illustration of a graph instance.

### 3.2. Fleet Placement Problem

The concept of a virtual station is created in this work, In the FPP model, carsharing stations are placed on the streets. A station is a virtual area on the city map defined by two elements, a center point, and a radius. Figure 2 illustrates this concept. A car is placed on a road which depicts a center point. A circle represents a radius of that center point. These two elements constitute the virtual station. A car can be picked up and returned to anywhere in that circle (station area). The coverage of a station is determined by the given maximum walking distance illustrated as a green line in the figure.

Due to the round-trip nature of the service, each car taken from the station needs to be returned to this station after completing the trip. The typical customers are people in residential areas (as shown in Figure 2) covered by the carsharing stations, who walk to pick up the nearest vehicle. Under the aforementioned assumptions and constraints, FPP can then be formulated.

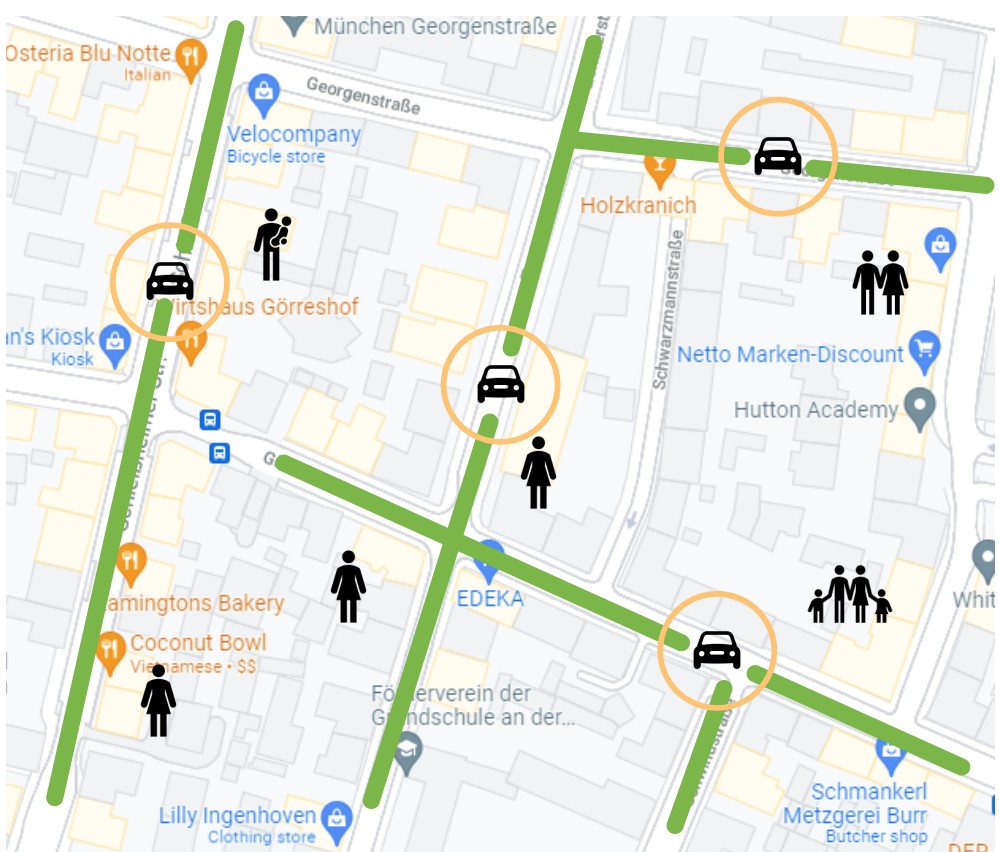

**Figure 2.** An illustration of a virtual area of a station on streets.

3.2.1. FPP Parameters

In the fleet placement problem (FPP), there are two types of parameters, inputs and instance parameters. Input parameters are the number of fleet stations, maximum walking distance, and station radius as shown in Table 1. Instance parameters are related to the problem graph instance, which are the set of street nodes, buildings, population, and distance between street nodes and buildings as summarized in Table 2. Stations are put on the street node in $S = \{s_1, s_2, \ldots, s_n\}$ and have a unique size $r$ (see Figure 2). Users are assumed to start the trip from their residence (referred to as building) in B = {b_1, b_2, ..., b_m} where each residence or building has a different number of users defined as $P = \{p_1, p_2, \ldots, p_m\}$. Each street node covers a set of buildings, which depends on the defined walking distance $w$ and a set of distance $D = \{d_{11}, d_{12}, \ldots, d_{ij}\}$ where $d_{ij}$ denotes walking distance between node $s_i$ and building $b_j$ where a distance function $d(s_i, b_j) = d_{ij}$. A station $s_i$ covers a building $b_j$ *iff* that station locates at most $w - r$ meters from the building.

**Table 1.** FPP input parameters.

| Input Parameters | Description | Type |
|---|---|---|
| $f$ | Maximum number of desired fleet stations. | $\mathbb{Z}^+$ |
| $w$ | Maximum walking distance allowed. | $\mathbb{R}^+_*$ |
| $r$ | Station area radius. | $\mathbb{R}^+_*, r \leq \frac{w}{2}$ |

**Table 2.** FPP instance parameters.

| Instance Parameters | Description | Type |
|---|---|---|
| $n$ | Number of street nodes. | $\mathbb{Z}^+$ |
| $m$ | Number of buildings. | $\mathbb{Z}^+$ |
| $i$ | Index for street nodes | $\mathbb{Z}^+$ |
| $j$ | Index for buildings | $\mathbb{Z}^+$ |
| $S$ | Set of street nodes (potential stations) | $S = \{s_1, s_2, \ldots, s_n\}$ |
| $B$ | Set of buildings (housing users) | $B = \{b_1, b_2, \ldots, b_m\}$ |
| $P$ | Set of population of buildings | $P = \{p_1, p_2, \ldots, p_m\}$. |
| $D$ | Set of walking distances between street nodes and buildings. | $D = \{d_{11}, d_{12}, \ldots, d_{ij}$ $: d_{ij} \in \mathbb{R}^+\}$ |

### 3.2.2. FPP Variables

The variable of FPP consists of three variables. The first variable is the set of the state of buildings $C$ where $c_i j$ represents the fact that building $b_j$ is or is not covered by a station $s_i$. If the building $b_j$ is covered then $c_{ij} = 1$ and $c_{ij} = 0$ otherwise. A building is covered when the center point of a station (at least) is located within walking distance smaller than $w - r$. The second variable is a set of state street nodes ($S'$), where $s_i'$ represents the state of node $s_i$ ($s_i \in S$). A node is active if it is the center point of a station. If $s_i$ is active then $s_i' = 1$ and $s_i' = 0$ if $s_i$ is inactive. It is important to note that a station can be active even if it does not cover any buildings. In this version, the station is able to accommodate only one vehicle at a time. Finally, the maximum global walking distance ($z$) denotes maximum walking distance from every selected stations to their covered buildings (hence, covered population) as shown in Table 3.

**Table 3.** FPP variables.

| Decision Variables | Description | Type |
|---|---|---|
| $C$ | Set of state of buildings | $C = \{c_{11}, c_{12}, \ldots, c_{nm}\}, c_{ij} \in \{0, 1\}$ |
| $S'$ | Set of state of street nodes | $S' = \{s_1', s_2', \ldots, s_m'\}, s_i' \in \{0, 1\}$ |
| $z$ | maximum global walking distance | $\mathbb{R}^+$ |

### 3.2.3. FPP Objectives

In the fleet placement problem, the objectives are to maximise the users coverage of a station and to minimise the maximum global walking distance between users and fleet stations. With these two objectives, the optimization model is formulated as follows:

$$\max \quad \sum_{i=1}^{n} \sum_{j=1}^{m} (c_{ij} \times p_j)$$

$$\min \quad z = max_{i,j}(s_i' \times c_{ij} \times d_{ij}) \qquad \forall i, j$$

$$\text{s.t.} \quad \sum_{i=1}^{n} s_i' \leq f \qquad\qquad\qquad\qquad (\text{C1})$$

$$(w - r)c_{ij} \geq s_i'(w - r) - (d_{ij} \times s_i') \quad \forall i, j \quad (\text{C2})$$

$$s_i' \geq c_{ij} \qquad\qquad\qquad\qquad \forall i, j \quad (\text{C3})$$

$$\sum_{i=1}^{n} c_{ij} = 1 \qquad\qquad\qquad\qquad \forall j \quad (\text{C4})$$

$$z \geq s_i' \times c_{ij} \times d_{ij} \qquad\qquad \forall i, j \quad (\text{C5})$$

$$z \geq 0 \qquad\qquad\qquad\qquad\qquad (\text{C6})$$

$$c_{ij} \in \{0, 1\} \qquad\qquad\qquad\quad \forall i, j \quad (\text{C7})$$

$$s_i' \in \{0, 1\} \qquad\qquad\qquad\qquad \forall i \quad (\text{C8})$$

$$i \in \{1, \ldots, n\} \qquad\qquad\qquad\quad (\text{C9})$$

$$j \in \{1, \ldots, m\} \qquad\qquad\qquad (\text{C10})$$

C denotes a constraint. Constraint 1 denotes that the number of stations cannot exceed the provided number $f$ of fleet stations. Constraint 2, 3, and 4 restrict the model to consider any building $b_j$ to be covered by one station and to be calculated only once for user coverage. The building $b_j$ is covered *iff* there is at least one active station in its proximity (the proximity is defined by $w - r$). Constraint 5 finds the maximum walking distance of the active station $s_i$ from all buildings $b_j$ that it covers. Constraint 6 impose the global walking distance to always be positive. Constraints 7 and 8 indicate that there are only two states for street nodes (active: 0, inactive: 1) and buildings (not covered: 0, covered: 1). Finally, constraints 9 and 10 denote the domains of indices $i$ and $j$ accordingly.

### 3.3. NP-Hardness Proof

According to Garey and Johnson [55], any decision problem that can be reduced from an NP-complete problem, whether it is a member of NP or not, is not solvable in polynomial time unless P = NP since it is as hard as the NP-complete problem. In order to prove the NP-hardness of FPP, its computational complexity is analyzed. Therefore, the decision counterpart of the fleet placement problem (FPP)–FPP–D is introduced. The decision counterpart FPP–D inherits all parameters from FPP.

In this section, the NP-hardness of FPP, through proving the NP-completeness of FPP–D, is demonstrated. For FPP–D, the question is to determine whether there exists a solution with $f$ station(s) such that all buildings are covered.

**Proposition 1.** *The FPP is NP-hard in the strong sense even if there is only one user in each building.*

**Proof.** We introduce a polynomial-time transformation to the FPP–D from the strongly NP-complete problem "Set Cover Problem (Minimum Cover Problem)" [55,56].

Set Cover Problem or SCP can be defined as follows: given a universe U of R elements, a collection of subsets of $U$, $G = \{g_1, g_2, g_3, ..., g_L\}$ and a positive integer $K \leq |G|$, the question is "Does G contain a cover for U of size K or less, i.e., a subset $G' \subset G$ with $|G'| \leq K$ such that every element of U belongs to at least one member of $G'$?"

Given an instance of SCP, we introduce the following instance of FPP–D. Firstly, let all buildings in $B$ be the equivalence of universe $U$ in SCP and $|B| = R$. Then, let S be the direct transformation of collection G where $S = \{s_1, s_2, s_3, ..., s_L\}$, such that $s_l = g_l$, $l = 1, 2, 3, \ldots, L$. In addition, we let $w, z = 1$ and $r = 0$ so that the building is covered if it is connected to the street node ($s_l$). With the prior assumption, the distances in matrix $D$ are assumed to be one if the street node is a 1-hop neighbour of the building and zero, otherwise. Therefore matrix $D$ reflects the membership of $S$ and is used to constitute the membership of collection $S$ in FPP–D. We also assume that there are L stations, hence $n = L$. Next, we assume $p_j = 1; j \in \{1, 2, 3, \ldots, |B|\}$ which means there is only user in building $j$. Let $c_j$ be one if a collection $s_i$ contains a building $b_j$ where $i \in \{1, 2, 3, \ldots, n\}$ and $j \in \{1, 2, 3, \ldots, |B|\}$. With the aforementioned assumptions, $\sum_{j=1}^{R}(c_j \times p_j) = |B|$. Finally, we let the threshold value $f = K$.

Let X be a solution to SCP. A solution for FPP–D is constructed in which the buildings in $B$ ($U$) are covered by $f$ stations where $s'_l = g_l \in X$, such that $x_l = s'_l$, if $s'_l \in X$ and $x_l = \emptyset$ if $s'_l \notin X$. Since X is a cover of $U$ (in SCP), all buildings in $B$ are covered and the number of stations in the corresponding solution (for FPP–D) is $f = |X|$.

Now assume that there exists a solution Y in FPP–D with $|Y| \leq K$ and $|Y|$ should not exceed $K$, otherwise, $|Y| > K$ and the condition will not hold. Therefore, there are at most $K$ station(s) with $y_i \neq \emptyset$. Since all buildings form $B$ and all buildings in $B$ belong to at least one member of $Y$, the selected stations with $y_i \neq \emptyset$ represents a solution to SCP, given a polynomial transformation from SCP to FPP-D. Since all input numbers in the FPP–D instance have a size most polynomial in the size of the input, FPP is strongly NP-hard. □

SCP (as an optimization problem) was proved to be polynomially non-approximable within the ratio $c \cdot \ln |G|$, for some constant $c > 0$ [57]. Therefore, we propose the following statement.

**Statement 1.** There exists no polynomial $(c \cdot \ln n)$- approximation algorithm for the FPP where $n$ is the input size, unless $P = NP$.

## 4. Optimization Methods

In this section, the state-of-the-art algorithms used in our experiments, PolySCIP [58], heuristics [5], and the Non-dominated Sorting Genetic Algorithm-II (NSGA-II) [45] are described. Each of them represents a different category of problem solver.

### 4.1. PolySCIP

The strength of exact algorithms is the guarantee of reaching the global optimum, but the related computational cost can prevent their usage for large size NP-hard problems. Examples of classical exact algorithms are branch and bound, branch and cut, or A*. There exist also commercial exact solvers such as IBM CPLEX [37] and AMPL [59] but to our knowledge, these algorithms and solvers are only able to solve single objective optimization problems. This limitation led to research for multi-objective exact solvers and in 2016, PolySCIP was proposed [58].

PolySCIP employs a "Lifted Weight Space Approach" [58]. This approach first optimizes the objectives lexicographically. The weighted (single objective) optimization problem from the first phase is optimized by using positive weight vectors. This guides the algorithm in exploring the Pareto front in the problem space. If the new non-dominated solution is found, the old solution (the one that has been dominated) is discarded and the process continues until all non-dominated solutions are found. As a result, the outcome is a Pareto front instead of just one solution. The method was proven mathematically in finding all global optima by the authors.

### 4.2. Heuristic Algorithms

Heuristic algorithms can be simply described as a set of rules to follow. A rule can be as simple as taking whatever that is the best in that particular instance (see Algorithm 1). Numbers of variations of heuristic algorithms were used in solving both MCLP and carsharing fleet management. Church and ReVelle [5] first proposed a heuristic algorithm, which adds one facility location at a time. The latter has also been applied to solve shared fleet placement problems [20,21] and is used in the comparative study in this work as well.

---

**Algorithm 1:** Greedy search algorithm

**Data:** Number of locations ($N$), Potential locations($L$)
**Result:** List of selected locations($S$)
1 $S \leftarrow \varnothing$
2 **for** $l \in L$ **do**
3 $\quad \mid \quad$ evaluate location $l$ using a fitness function
4 **end**
5 sort locations according to fitness score
6 $S \leftarrow N$ best fitness location ($l$)
7 Return $S$

---

This heuristic algorithm relies on an iterative search. The algorithm starts with an empty list of locations. Then, in each iteration, each location is evaluated according to the fitness function. The algorithm then adds the location with the highest fitness score in the list and that selected location is removed from the location pool, in order not to be re-selected in the subsequent iteration. This algorithm has a complexity of $O(sn)$ where $s$ is the number of (desired) stations and $n$ is the number of street nodes. The pseudocode of the algorithm is shown in Algorithm 2.

---

**Algorithm 2:** Iterative search algorithm

---

**Data:** Number of locations (*N*), Potential locations(*L*)

**Result:** List of selected locations(*S*)

1   $S \leftarrow \varnothing$

2   **for** $n \in \{1 \ldots N\}$ **do**

3      **for** *location(l)* **in** *L* **do**

4         evaluate location *l* using a fitness function

5      **end**

6      sort location according to fitness score

7      $S \leftarrow$ best fitness location

8      remove facilities that are covered by location *l*

9   **end**

10   Return *S*

---

Even though the algorithm was first introduced to solve single objective optimization problems, there are several ways to adapt it to solve multi-objective problems, e.g., weighted sum, and $\epsilon$-constraint. These variations can be extended further to yield an approximated front as a result. Using a weighted sum approach, each objective fitness is normalized as shown in Equation (1).

$$F_n = \frac{F_a - LB}{UB - LB},$$

(1)

where $F_n$ is the normalised fitness and $F_a$ is the actual fitness (e.g., coverage, walking distance, or bi-objective) before normalization. $UB$ is the upper bound (the highest fitness) and $LB$ is the lower bound (the lowest fitness). With this weighted sum approach, the priority of each objective can be adjusted as shown in Equation (2). The iterative search can be launched multiple times with different weight ratios for each optimization objective. Let us for instance assume that there is the following list of weights: [(0.1,0.8,0.1), (0.33, 0.33, 0.33), (0.1, 0.1, 0.8)]. With these weights, up to three solutions can be reached.

$$F_t = \sum_{i=1}^{m} w_i F_i$$

(2)

where $F_t$ is the total fitness score from the weighted sum and $w_i$ is the weight associated to the objective $i$. Finally, $F_i$ is the normalized fitness score of objective $i$. The granularity of weights can also be adjusted at the cost of a higher computation cost since more combinations of weights require more executions of the algorithm. Once a certain amount of solutions (decided by the decision maker) is collected, an approximated front can be constructed.

In this work, we study six variants of heuristic algorithms based on the greedy and iterative versions. They are;

1. Coverage-focused greedy algorithm;
2. Distance-focused greedy algorithm;
3. Bi-objective-focused greedy algorithm;
4. Coverage-focused iterative algorithm;
5. Distance-focused iterative algorithm;
6. Bi-objective-focused iterative algorithm.

The purpose is to establish a baseline for comparison and to evaluate the performance of the state-of-the-art heuristic algorithms against other algorithms.

### 4.3. Non-Dominated Sorting Genetic Algorithm-II (NSGA-II)

In the Evolutionary Algorithm (EA) approach to solving a problem, a well known concept called 'metaheuristic' can be concisely defined as a higher-level procedure or strategy for a partial search. Hence, a global optimum is not guaranteed, but it generally yields acceptable results. Metaheuristics usually contain a stochastic process, which make

them non-deterministic. NSGA-II is a metaheuristic optimization algorithm that is based on Pareto-dominance [45]. Pareto-dominance is defined as follows:

$$z \succ z' \Leftrightarrow \forall i \in \{1, 2, \ldots, n\}, z_i \leq z'_i \cup$$
$$\exists j \in \{1, 2, \ldots, n\}, z_j < z'_j ,$$

where $z$ and $z'$ are vectors of objectives in $Z$ and $z \succ z'$ means $z$ dominates $z'$. If the selected solutions are both non-dominated, one of the parent solutions is selected at uniformly random. Other metaheuristic algorithms in this category are Simple Evolution Algorithm for multi-objective Optimization (SEAMO) [60], Strength Pareto Evolutionary Algorithm II (SPEA2) [46], and Pareto Envelope-based Selection Algorithm (PESA) [61]. NSGA-II was shown to be more efficient than the previously mentioned algorithms in a GSM antenna location problem (a variant of FLP) [17].

Among these algorithms, NSGA-II is renowned due to its numerous proven applications. It is largely based on the Genetic Algorithm (GA), starting from population initialization, selection of parents, crossover, and mutation to obtain a new population of solutions. Individuals in both the parent and offspring populations are sorted according to their rank, and the best solutions are chosen to create a new population. If individuals have the same rank, a density estimation based on the crowding distance to the surrounding individuals of the same rank is used. A new reference-point-based NSGA-II called NSGA-III is proposed, with the intention on solving problems with three or more objectives [62]. Hence, in this work, NSGA-II is selected as the problem is bi-objective. The pseudocode of NSGA-II is shown in Algorithm 3.

---

**Algorithm 3:** Nondominated Sorting Genetic Algorithm (NSGA-II)

1 population ← InitializePopulation (size)
2 Evaluate (population)
3 NondominatedSort (population)
4 CrowdingDistance (population)
5 **while** *termination criteria are not yet satisfied* **do**
6     parents ← TournamentDCD (population)
7     offspring ← recombination+mutation(parents)
8     Evaluate (offspring)
9     NondominatedSort (population + offspring)
10     CrowdingDistance (population + offspring)
11     population ← CrowdedComparison (population + offspring)
12 **end**
13 Return population

---

Next, the solution encoding population initialization and evolutionary operators employed in NSGA-II to solve the fleet placement problem are presented in detail. More details on NSGA-II's specific operators can be found in [45].

**Solution Encoding:** A solution is a string of fleet locations denoted by 'id numbers' from Openstreetmap. An example is shown in Figure 3, blue dots are candidate fleet locations on the streets. The shown numbers are ids from Openstreetmap. An example solution contains five locations (location number 1 to 5). Naturally, it is one of many possibilities in this example area. This encoding is more suitable than the binary encoding due to the size of the problem instances, which can be exceedingly large (more than 100,000 street nodes is possible in reality). The encoding contains no order and swapping genes in the chromosome does not change the fitness of the solution.

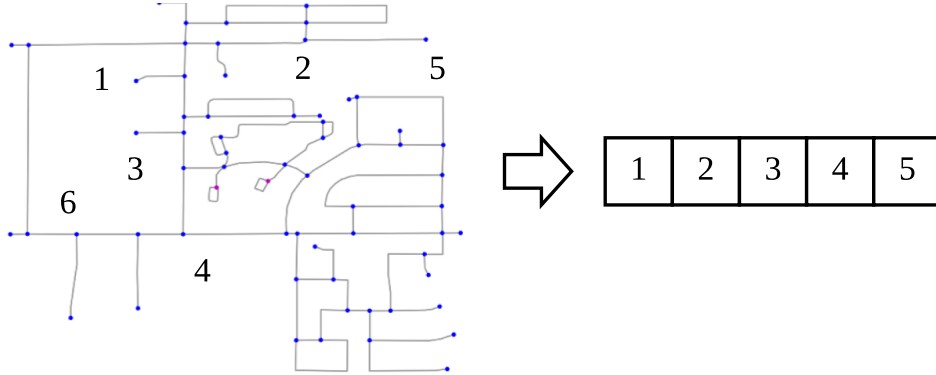

**Figure 3.** Example of a chromosome representing a possible solution. Street nodes are represented by IDs (numbers) and a solution is consisted of these IDs.

**Population Initialization:** A solution is initialized by randomly choosing (based on a uniform distribution law) street nodes from all street nodes in a problem instance. Note that each street node in the algorithm may at most be selected once in each solution during the initialization.

**Crossover:** A two-point crossover is adopted in this work. The process randomly selects two points in both solutions as starting and ending points for exchanging portions and recombines these portions to create two new solutions as shown in Figure 4. Although, this crossover process may introduce solutions with redundant placements, due to fitness score calculation (e.g., redundant locations lead to lower coverage) those solutions will be deemed as low quality and hence be eliminated in the next generation of selection. This reduces the execution time of the algorithm.

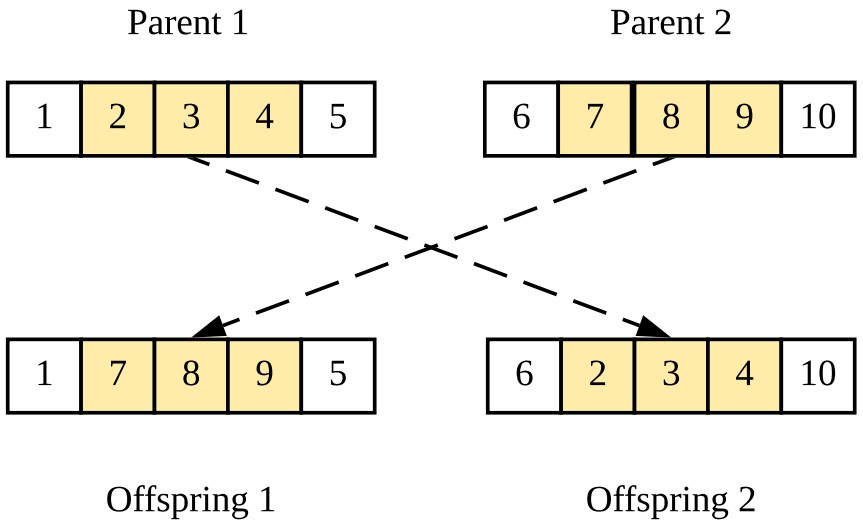

**Figure 4.** Two-point crossover process.

**Mutation:** The uniform mutation operates a replacement coming from the pool of all vehicle locations defined by street node IDs. Figure 5 illustrates the mutation operator where *Sample* is a function to randomly pick one location from the pool and $1, 2, 3, \ldots, n$ denotes all street node IDs. However, if the replacement already exists in the current solution, the process is repeated until a valid replacement is found. As mentioned before in crossover, redundant solutions will be eliminated by the process.

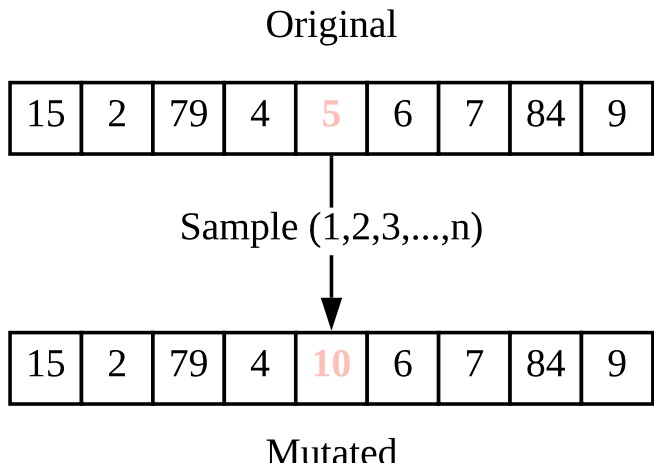

**Figure 5.** Uniform mutation process.

It is worth noting that evolutionary operators inherit known shortcomings such as local optimal and plateau. Evolutionary operators in this work are no exception. Several combinations of crossover and mutations have been experimented and none resulted in guaranteed superiority over all others. This work focuses on the implementation of the FPP model rather than finding suitable parameters and operators for the metaheuristic algorithm (i.e., NSGA-II) under consideration. Therefore, it may be possible that adjustment and tuning of evolutionary operators for their suitability may be necessary for its application in some instances that share too few characteristics with instances in this work.

### 4.4. Multi-Objective Performance Metrics

Several multi-objective quality metrics exist, which can be categorized based on the quality aspect that they assess, i.e., convergence (distance to the optima), diversity, and both convergence and diversity altogether [33]. In this work, we consider the three commonly used three indicators which measure the complementary aspects of the yielded solutions, namely, Inverted Generational Distance (IGD) [63], Spread [45] and Hypervolume (HV) [33]. Since the exact Pareto front can only be computed on small size instances, for large instances a reference front is obtained by combining the approximated Pareto fronts resulting from the heuristic and metaheuristic algorithms.

**Inverted Generational Distance (IGD)** [63]: This metric measures the distance between the obtained approximated solutions and the Pareto front. IGD is defined in Equation (3), where $d_i$ is the Euclidean distance from point $i$ in the approximated front to the closest one in the Pareto front, and $n$ is the number of solutions in the Pareto front. $IGD = 0$ indicates that the evaluated Pareto front consists only of solutions from the optimal Pareto front.

$$IGD = \frac{\sqrt{\sum_{i=1}^{k} d_i^2}}{n} \quad . \tag{3}$$

**Spread** [45]: This metric measures the diversity of the obtained approximated front and is defined as:

$$\blacksquare = \frac{d_f + d_l + \sum_{i=1}^{N-1} |d_i - \bar{d}|}{d_f + d_l + (N-1)\bar{d}} \quad , \tag{4}$$

where $d_i$ is the Euclidean distance between consecutive solutions, $\bar{d}$ is the mean of these distances, and $d_f$ and $d_l$ are the Euclidean distances to the *extreme* solutions of the Pareto front. A zero value indicates an ideal distribution, i.e., pointing out a perfect spread of the

solutions in the evaluated set of solutions.

$$HV = \text{volume}\left(\bigcup_{i=1}^{|Q|} v_i\right).\tag{5}$$

**Hypervolume** [64]: This metric assesses both convergence and diversity of a Pareto front. It calculates the m-dimensional volume (in the objective space) covered by the solutions in the evaluated Pareto front $Q$ and a dominated reference point $W$. For each solution $i \in Q$, a hypercube $v_i$ is constructed with the reference point $W$ and the solution $i$ as the diagonal corners of the hypercube. The hypervolume is calculated as the union of all hypercubes, as shown in Equation (5). The higher the hypervolume the better the algorithm performed.

## 5. Execution of the Proposed Model

This section is composed of three parts. The first two parts present the process of building graphs from real street maps and the demographic data integration into the graphs. The third part describes the parameters and environment the execution.

### 5.1. Building Graph Instances

Most of the public street maps are usually not available in a graph format. To build graphs, street map data and footprints of the buildings are obtained from "OpenStreetMap" [65] using "OSMnx" [66] and "NetworkX" [67] tools. Then, a simplified graph is created, it combines a street map and buildings together with respect to the real position of the street nodes and buildings in $V$. Each edge between a building and its nearest street node is built using Open Source Routing Machine (OSRM) [54]. Edges are assigned a weight that is proportional to the walking distance extracted from OSRM. An example of a resulting instance is shown in Figure 6. Gray edges represent streets. The nodes on the streets represent street nodes, while the nodes outside the streets represent buildings in the area. Input data and parameters for all instances are summarized in Table 4.

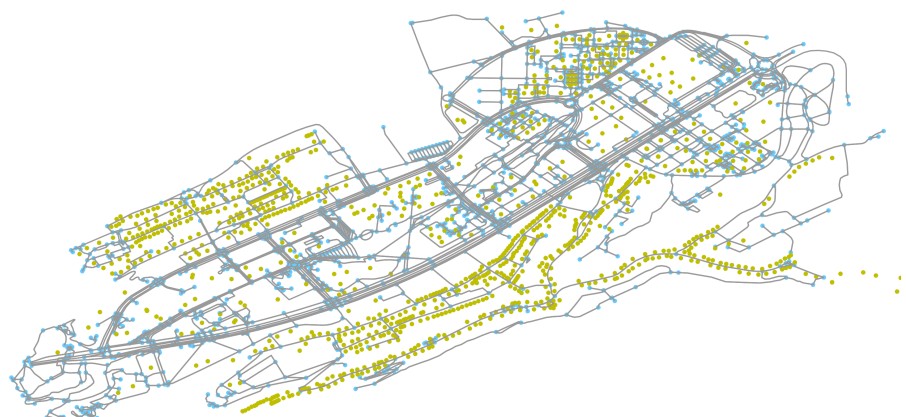

**Figure 6.** LU2 instance. Blue nodes represent possible locations for carsharing stations. Yellow nodes represent residential buildings.

### 5.2. Problem Instances

The performance of PolySCIP, heuristic algorithms (simple and iterative) and NSGA-II algorithms is compared on three real instances; LU1, LU2, and MU1 as shown in Table 4. LU1 is a small portion of Luxembourg city containing 63 street nodes and 47 buildings. LU2 consists of 2 districts of Luxembourg city that contain a total of 2026 street nodes and 1063 buildings. MU1 is an inner part of the city of Munich, which contains 16,075 street nodes and 21,816 buildings.

For the LU1 instance, the radius of the carsharing station ($r$) is set to zero and the maximum walking distance for users ($w$) is 150 m for all evaluated algorithms. The number

of stations is set to four. The LU1 setup is ideal for observing and understanding the operational details of the evaluated algorithms.

The size of the LU2 graph reflects two city districts and is the smallest portion for real-world planning (deemed by the business expert). The maximum acceptable walking distance depends on the selected mode of transportation. Daniels and Mulley [53] show that people are willing to walk significantly longer to take a train than a bus as long as they deem it worthy. For carsharing, a realistic walking-to-the-car distance ($w$) is around 500 m [68]. The population is selected to match the real numbers reported by the city of Luxembourg [69] and is distributed uniformly in residential buildings. The area covered by a carsharing station has a radius ($r$) of 100 m, while the number of stations is set to 10.

The MU1 instance aims to evaluate the performance of algorithms on a city-wide scale. The population in each district is taken from municipalities and is distributed uniformly in residential buildings . We estimated the number of carsharing users at 2% of the total population. The station radius ($r$) and walking distance ($w$) are the same as in the LU2 instance, but the number of stations is increased to 100 stations. In all three instances, carsharing users are distributed uniformly among available buildings on the map. All of these settings have been defined by domain experts based on the study and real deployment plan.

**Table 4.** Problem instances.

|  | LU1 | LU2 | MU1 |
|---|---|---|---|
| City | Luxembourg | Luxembourg | Munich |
| Population | 561 | 11,439 | 17,486 |
| Number of carsharing stations ($f$) | 4 | 10 | 10,072 |
| Number of street nodes | 63 | 2026 | 16,075 |
| Number of residential buildings | 47 | 1063 | 21,816 |
| Maximum walking distance ($w$) | 150 m | 500 m | 500 m |
| Carsharing station area radius ($r$) | 0 m | 100 m | 100 m |

*5.3. Algorithms Implementation and Parameters*

PolySCIP (version 4.0) and heuristic algorithms are deterministic and do not require any parameters apart from those shown in Table 4. As NSGA-II is evolutionary based, it requires initial population and parameters for GA operators. Table 4 reveals the values adopted for these three instances. NSGA-II is executed for 30 times due to its stochastic nature. We develop heuristic algorithms and NSGA-II using Python 3.7 and DEAP (a library for metaheuristic algorithms) [70]. It is a common practice to include seed solutions in the initial population of metaheuristic algorithms. In this work, these are injected into the initial population as a seed solution from coverage-focused and distance-focused iterative heuristic algorithms. The configurations for NSGA-II are mentioned in Table 5.

**Table 5.** NSGA-II configuration parameters.

|  | LU1 Instance | LU2 Instance | MU1 Instance |
|---|---|---|---|
| Number of generations | 400 | 400 | 400 |
| Population size | 20 | 50 | 100 |
| Selection process | Tournament | Tournament | Tournament |
| Crossover method | 2-point crossover | 2-point crossover | 2-point crossover |
| Crossover rate | 0.8 | 0.9 | 0.9 |
| Mutation rate | 0.01 | $\frac{1}{\#stations}$ | $\frac{1}{\#stations}$ |

## 6. Results

The results of each instance (LU1, LU2 and MU1) are represented as a scatter plot where the x-axis represents the maximum walking distance (lower is better) and the y-

axis represents the number of covered users (higher is better). Execution in this work is performed on a single core of an Intel Xeon L5640 (2.26 GHz) with courtesy of University of Luxembourg HPC.

### 6.1. Result of LU1 Instance

Figure 7 presents the obtained Pareto fronts from eight different algorithms and Table 6 provides the numerical results obtained for all evaluated algorithms. To simplify the table, only two extreme points on both Pareto fronts are shown. The highest achieved coverage is 391 users, which is yielded by the iterative heuristic algorithm, the exact method, and NSGA-II. On the other hand, the lowest distance of 93.5 m is achieved using PolySCIP.

**Table 6.** Numerical results in LU1 instance. Only extreme solutions from the two Pareto fronts are mentioned.

|  | Covered Users | Maximum Walking Distance (Meters) |
|---|---|---|
| PolySCIP (Best coverage) | 391 | 149.528 |
| PolySCIP (Best distance) | 108 | 93.546 |
| NSGA-II (Best coverage) | 391 | 149.528 |
| NSGA-II (Best distance) | 203 | 106.4 |
| Coverage Heuristic | 348 | 148.491 |
| Distance Heuristic | 187 | 112.398 |
| Bi-objective Heuristic | 333 | 144.515 |
| Coverage Iterative Heuristic | 391 | 149.528 |
| Distance Iterative Heuristic | 87 | 106.4 |
| Bi-objective Iterative Heuristic | 358 | 144.401 |

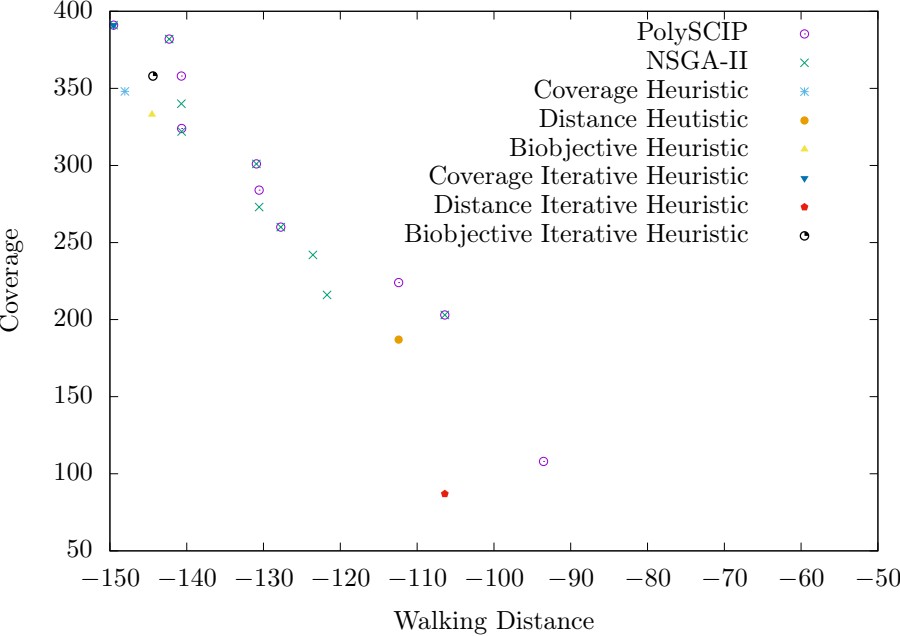

**Figure 7.** Results in the validation phase. The higher the value on the x axis (moving toward the right), the better the distance objective is. The higher the value on y axis, the better the user coverage objective is.

Overall, results from NSGA-II are debatably superior when both objectives are considered. The results are close to the optimum (106.4 from NSGA-II and distance-focused iterative heuristic and 93.546 m from PolySCIP). In fact, the distance result from NSGA-II is even better than its iterative counterpart as it covers more users. Figure 8 shows the stations and their respective coverage in LU1 instance.

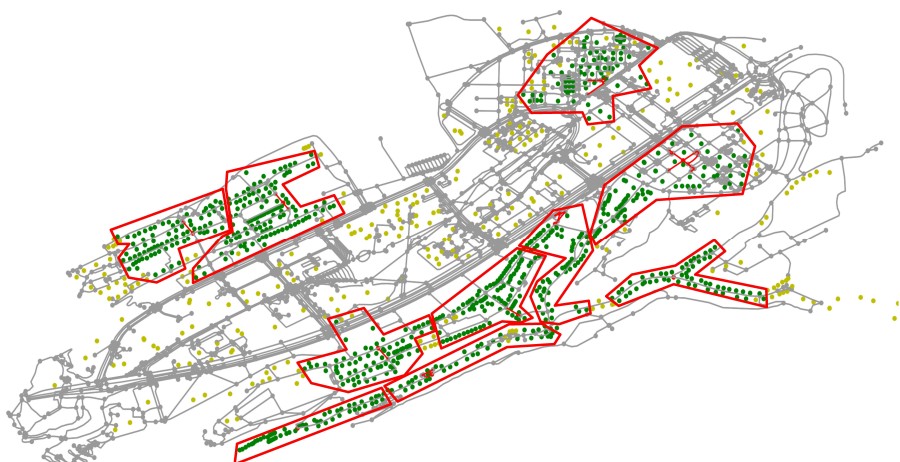

**Figure 8.** A map showing a solution from NSGA-II that yields the highest user coverage and the highest maximum walking distance. Covered buildings are depicted as nodes inside polygons.

For the IGD indicator (see Table 7), NSGA-II yields 3.02. This value is in accordance to Figure 7 between a true Pareto and an approximated fronts. It can be seen that NSGA-II achieved some of the solutions on the true Pareto front—especially the highest user coverage solution.

**Table 7.** Comparing Pareto fronts for PolySCIP and NSGA-II.

|              | **IGD**            | **Spread** | **HV** |
| ------------ | ------------------ | ---------- | ------ |
| Exact method | True Pareto front  | 0.488      | 0.449  |
| NSGA-II      | 3.02               | 0.525      | 0.351  |

As for the spread indicator, the true Pareto front yields 0.488, while the NSGA-II Pareto front yields 0.525. This means the diversity in the exact method Pareto front is better than NSGA-II's. This was due to the fact that the coverage objective overwhelmed the distance objective leading to a cluster of solutions in the upper right region in Figure 7. The hypervolume of the true Pareto front is 0.449 and the NSGA-II Pareto front yields 0.351. The difference occurs because some solutions of NSGA-II are dominated by PolySCIP's.

It is essential to note that PolySCIP was applicable for the LU1 instance due to its small size. However due to the FPP complexity, for larger instances like LU2 and MU1, PolySCIP became an enviable approach. This is elaborated in Sections 6.2 and 6.3.

### 6.2. Result of LU2 Instance

There is only one Pareto front from NSGA-II in Figure 9 since PolySCIP cannot deliver the solutions even after 18 days and has a memory usage of 84 GB. The plot shows that NSGA-II yields a higher coverage than the iterative heuristic coverage algorithm when it takes the iterative methods' solutions as seed solutions in the initial populations. The highest achieved user coverage is 8421 users with a maximum walking distance of 399.8 m. On the other hand, the lowest maximum walking distance achieved is 135.7 m with only 47 covered users. In Table 8, the best results from each category (i.e., simple heuristic, iterative heuristic, and NSGA-II), are compared. It reveals that NSGA-II achieves the highest user coverage and lowest walking distance among all algorithms. It can be observed in Figure 9 that even though some residential buildings are located close to carsharing stations, they are not covered. This is because the entrances of those buildings (determined during the snapping process) are mapped on the opposite streets, which are not covered by the stations. However, the number of such buildings is marginal and can be neglected.

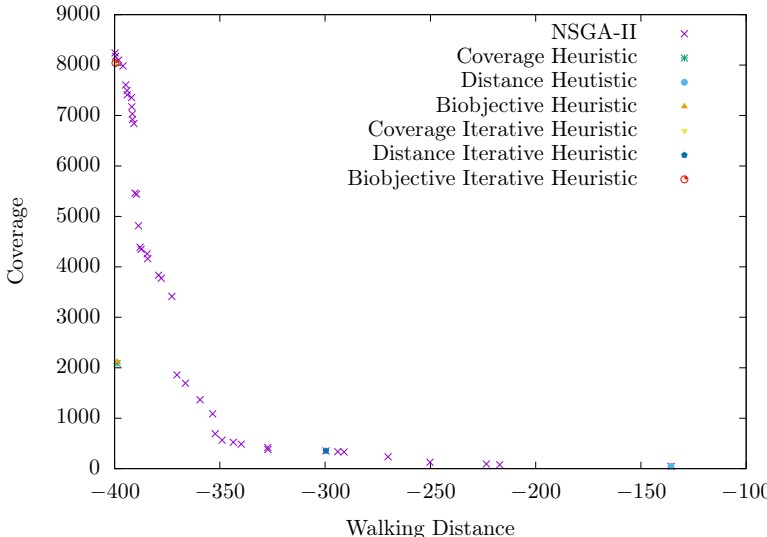

**Figure 9.** NSGA-II's Pareto front and heuristic algorithms' solutions for LU2. The higher the value on the x axis (moving towards the right), the better the distance objective is. The higher the value on the y axis, the better the user coverage objective is.

**Table 8.** Comparing best results from each algorithm categories in LU2 instance.

| Algorithm | Coverage Oriented | | Distance Oriented | |
|---|---|---|---|---|
| | Covered Users | Walking Distance | Covered Users | Walking Distance |
| Simple Heuristic | 2100 | 399.8 | 47 | 135.7 |
| Iterative Heuristic | 47 | 135.7 | 231 | 300 |
| NSGA-II | 8421 | 399.8 | 47 | 135.7 |

### 6.3. Result of MU1 Instance

Due to the larger input size of the MU1 instance and FPP being NP-hard, PolySCIP cannot be employed. It takes 17 h to come up with one solution for iterative heuristic algorithms, while it takes 26 min for NSGA-II to come up with an estimated front (read Table 9). Their respective results are presented in Figure 10. The obtained results are consistent with LU1 and LU2 instances. Table 9 presents the execution time of all algorithms. Although simple heuristic algorithms take only 7 min to find a solution, the results are not comparable to the others, which are more complex. From the results, NSGA-II also achieves higher user coverage and shorter walking distance than the heuristics.

**Table 9.** Execution time for NSGA-II and heuristic algorithms on MU1. The measured time depicts the execution time each algorithm takes to locate 100 stations.

| Algorithm | Execution Time |
|---|---|
| NSGA-II | 26 min |
| Coverage Heuristic | 7 min |
| Distance Heuristic | 7 min |
| Bi-objective Heuristic | 7 min |
| Coverage Iterative Heuristic | 17 h |
| Distance Iterative Heuristic | 17 h |
| Bi-objective Iterative Heuristic | 17 h |

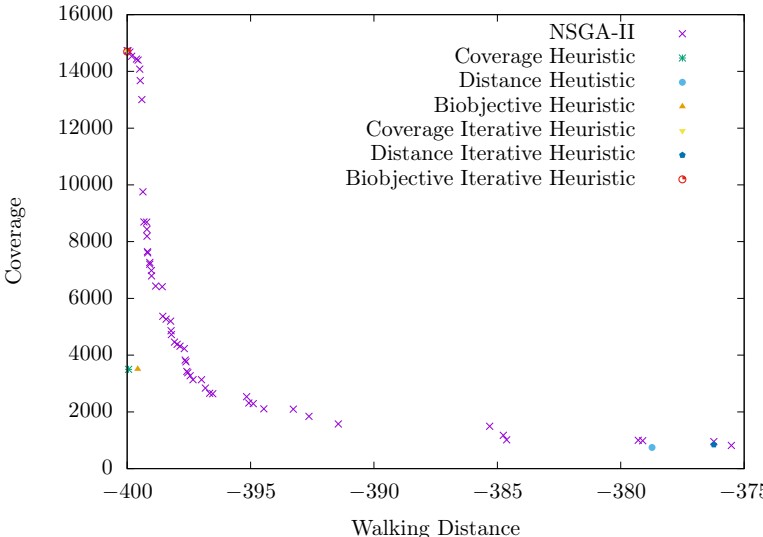

**Figure 10.** NSGA-II Pareto front and solutions of heuristic algorithms for the MU1 instance.

Figure 11 illustrates the NSGA-II fleet placement solution, which maximizes the number of covered users. Red pins mark locations of the carsharing stations, while green polygons show designated parking areas in inner Munich. All heuristic algorithms and NSGA-II are also compared using 72 stations in MU1 instance to compare with the manual allocation. Figure 12 shows that the manual allocation is of a lesser quality than some of the (meta-)heuristic algorithms, the iterative coverage and bi-objective version in particular, and NSGA-II. The comparison between the best results from each category of algorithm and manual allocation is also shown in Table 10. The difference in results in terms of user coverage is up to 50% (manual allocation being on the lower end), while the walking distance is similar. The results also further stress the benefit of Pareto front in decision making since it offers more options to choose from compared to the heuristic algorithms.

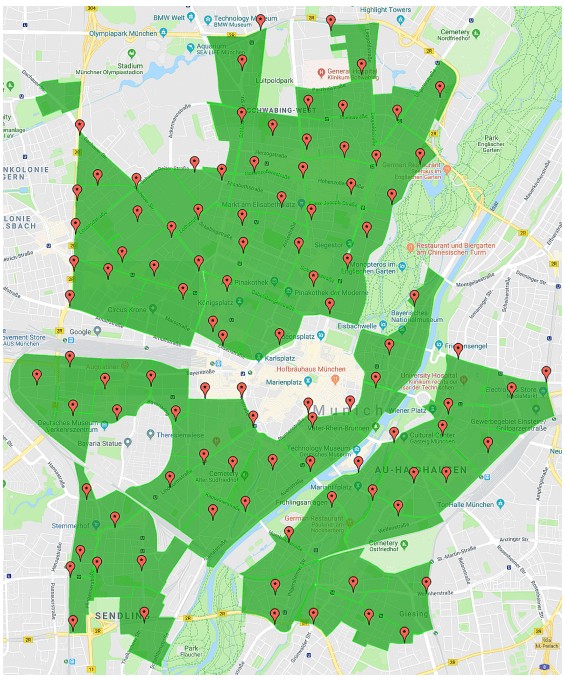

**Figure 11.** NSGA-II fleet placement solution which maximises user coverage in the city of Munich. Red pins are locations of the carsharing station. Green zones indicate the inner area of Munich.

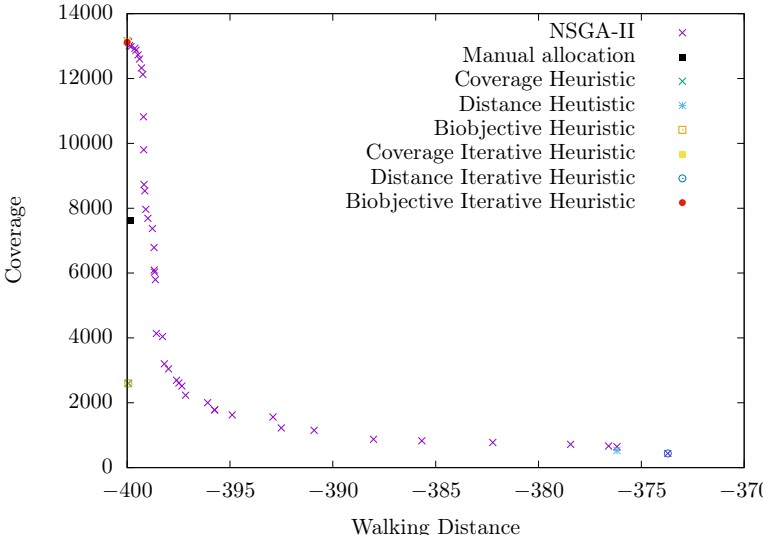

**Figure 12.** NSGA-II approximated Pareto front and solutions of heuristic algorithms in MU1 instance compared to the manual allocation (72 stations).

From the MU1 instance results, iterative heuristic approaches may still be possible but usually at the expense of extremely high computation time. Moreover, discovering suitable heuristics is problematic in its own. This is where the use of metaheuristic algorithms (e.g., NSGA-II) is proven to be more effective in term of solution qualities and computation time.

**Table 10.** Comparing the best results from each algorithm categories and manual allocation in MU1 instance.

| Instance | Algorithm | Coverage Oriented | | Distance Oriented | |
|---|---|---|---|---|---|
| | | Covered Users | Walking Distance | Covered Users | Access Distance |
| Simulation | Simple Heuristic | 3421 | 399.4 | 1091 | 378.6 |
| | Iterative Heuristic | 14,892 | 399.8 | 1214 | 376.1 |
| | NSGA-II | 14,892 | 399.8 | 1256 | 375.2 |
| Real-world | Simple Heuristic | 2291 | 399.8 | 1124 | 375.9 |
| | Iterative Heuristic | 13,224 | 399.8 | 986 | 374.2 |
| | NSGA-II | 13,224 | 399.8 | 986 | 374.2 |
| | Manual Allocation | 7864 | 399.8 | 7864 | 399.8 |

*6.4. Discussion*

**Efficiency and performance:** When optimality is of real concern, PolySCIP is best applied. However, its high execution time and memory requirements make it inapplicable in practice where the instance is anything greater than 1390 street nodes and 1063 buildings. PolySCIP was unable to find a solution at an acceptable time (unfinished even after 18 days), and these numbers do not represent anything close to the size of a typical large city.

Greedy algorithms, on the other hand, have the lowest execution time of all, but their results are unacceptable for practical applications due to their low user coverage. The underperformance of basic heuristic algorithms is alleviated in the iterative version, however, it comes with an additional computation cost. Despite the low execution time for a small instance, it becomes an issue in a larger instance. In the MU1 instance, the simulations took 17 h to locate 100 carsharing stations (on Intel Xeon L5640 at 2.26 GHz and over 128 GB or memory).Increasing the number of stations or increasing the size of the analyzed area will increase the execution time (in a factorial term, $n!$, where n is a number of locations) and can make it impractical.

NSGA-II's main advantages are the approximated front, the ability to cope with the size of problem instance, and the ability to improve existing solutions even further if possible. NSGA-II is 30 times faster than iterative algorithms and is still able to produce alternative solutions without needing to rerun the algorithm and change weights (in a

bi-objective iterative algorithm). These properties make NSGA-II an attractive choice in finding applicable fleet placement solutions, additionally it yields a better quality solution in term of coverage than the manual allocation, which usually takes a much longer execution time.

**Coverage vs. walking distance:** After observing the coverage quality of distance-oriented algorithms, we found that the after the minimization of walking distance, the distance is only marginally reduced. Further analysis was carried out on the MU1 instance, with an equal weight of 0.5 to both objectives for bi-objective-focused iterative algorithm. The solution should be at $(-397, 3000)$ in the approximated Pareto front in Figure 10. However, it is located at $(-399, 14{,}700)$ instead. There are also solutions where the walking distance and user coverage are low, this is because they place stations away from crowded areas and situate the stations near a few buildings, hence, claiming the low maximum walking distance and yield lower user coverage (see Figure 13). The decrease in walking distance (in those solutions) only translates to a two to three minutes difference on foot.

The marginal difference in walking distance can be explained by the nature of the city. Users are clustered in a densely populated area. If a carsharing station is placed in such an environment, there would be users at the edge of the coverage, which makes the maximum walking distance for users to be as high as the maximum coverage distance. A station can be relocated to lower the walking distance, but the user coverage is also likely to be lower in the process. On the other hand, the walking distance can be drastically low if carsharing stations are located in an uninhabited area, but this would be detrimental to the user coverage objective and contradicts the main purpose of a carsharing service. Hence, in this work, we have shown the effect of walking distance objective in carsharing fleet placement.

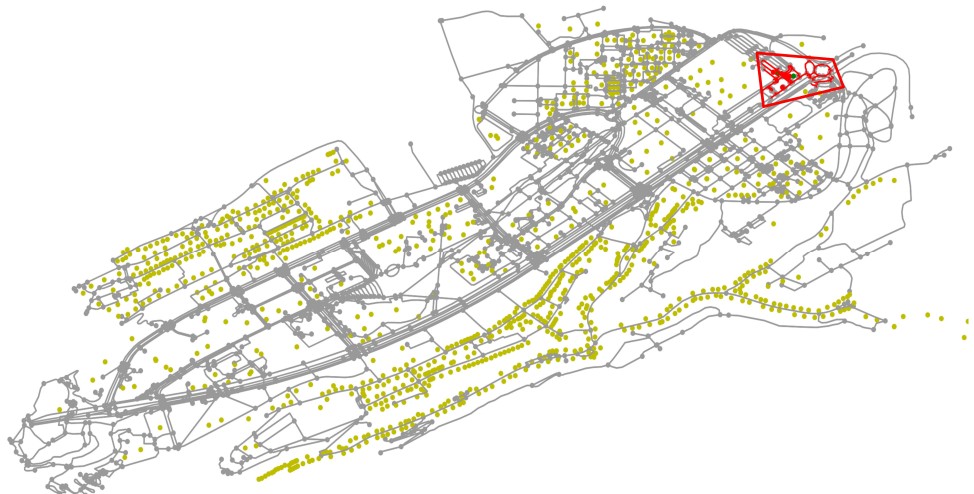

**Figure 13.** A solution that yields a low global walking distance, but also yields low user coverage.

## 7. Conclusions

This work proves that realistic Fleet Management Problem is an NP-hard problem, Apart from suggesting that exact and optimal solutions may not be realizable, it paves way to the application of heuristic and metaheuristic algorithms. This work proposed a novel methodology in optimizing fleet placement in station-based round-trip carsharing and suggests how such problems can be modeled. It is among the first to model the problem of fleet placement in carsharing and to apply a state-of-the-art optimization algorithm in attempting to determine satisfactory solutions. A set of heuristic, metaheuristic (NSGA-II), and exact (multi-criteria solver) algorithms have been applied and their performance evaluated on three instances with two objectives, i.e.,maximizing the number of carsharing users and minimizing the maximum global walking-to-the-car distance, under consideration.

This work is also the first to use real and exact instances (instead of just an abstract) for the study. Three different instances have been used, and each has its own characteristics with different sizes and objectives to reflect real world demand. The proposed method demonstrates that NSGA-II is superior to the manual allocation by a significant margin in user coverage and in terms of approximated Pareto front, and presents a number of solutions for decision makers to choose from. Solutions from our proposed method are also more efficient in terms of both user coverage and walking distance. While a metaheuristic approach has received much attention lately, this works affirms its application in transportation and Fleet Management Problem, in particular. The model proposed ought to be a good starting point in solving similar problems for further research among transportation and logistic communities.

Future work could apply the proposed approach to other cities such as London, Athens, and Paris. Additional objectives such as car fleet utilization, car fleet size, and the number of stations may also be included, since these are also real concerns after the initial launch of the carsharing business. A tailor-made metaheuristic algorithm may also be invented with the Fleet Management problem in mind too.

**Author Contributions:** Conceptualization, B.C., D.K. and P.B.; methodology, B.C., Grégoire Danoy, M.B. and F.G.; software, B.C.; validation, B.C., J.M. and K.L.; formal analysis, B.C., F.G. and J.M.; investigation, B.C. and D.K.; resources, D.K. and P.B.; data curation, B.C.; writing—original draft preparation, B.C., G.D. and D.K.; writing—review and editing, B.C. and K.L.; visualization, B.C.; supervision, P.B. All authors have read and agreed to the published version of the manuscript.

**Funding:** This research received no external funding.

**Institutional Review Board Statement:** Not applicable.

**Informed Consent Statement:** Not applicable.

**Data Availability Statement:** Not applicable.

**Conflicts of Interest:** The authors declare no conflict of interest.

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
