# Peer review of "Optimization of Carsharing Fleet Placement in Round-Trip Carsharing Service"

_applsci, doi:10.3390/app112311393_

Round 1
Reviewer 1 Report
This article proposes a round-trip carsharing system in which the initial location of the fleet's vehicles is optimised. Each shared car must be parked in a specific location that is easily accessible to the majority of residents. The Fleet Placement Problem (FPP) is a unique optimisation model for the round-trip carsharing fleet placement problem that solves the problem utilising three separate optimisation algorithms: PolySCIP (exact approach), heuristics, and NSGA-II (metaheuristic). Their results are compared using two real-world examples, Luxembourg and Munich.
The topic of the paper is interesting, but some concerns should be addressed. The main one is the problem statement, section 3. This section is not well-structured and is very hard to read. I strongly suggest the authors entirely revise this section. In addition, there are some more comments that should be addressed:
- The title of the paper can be defined better. “Automatization of Carsharing” not only is a bit unclear but also, I think is off-topic. I had a look at some papers by the first author. I think this paper has been extracted from his thesis. Other papers by these authors on this topic have more relevant titles in comparison with the title of this paper. My suggestion is to revise it.
- The research problem is a bit unclear in the abstract. In addition, this is an optimization problem, the objective should be clarified in the abstract.
- In line 34 “As a result, station-based services are less flexible, with an advantage of easier implementation and management.” This sentence can be considered as a good motivation for this study. Please elaborate it more and bring example(s) if it is possible.
- In line 37, “In this work, a new method to automatize the fleet placement……” this word “automatize” has been repeated many times in the paper, but I believe it is not relevant to the paper at all. This study is an optimization problem and nothing else.
- Figure 1 and its explanations in the body of the paper is unclear. I believe this figure can be presented in a better way.
- Section 3, particularly, Section 3.1 and 3.2 are very unclear. please see similar papers in this research area. The most common way is to bring a research problem (or statement) and explain the problem clearly (preferably using figures). Then bring one or several tables and list the parameters, set, decision variables, and then present the model. In addition, the proposed model has some problems, for example, types of decision variables have not been defined (continues, binary, ….) in addition the model C2 should be modelled in a proper way. You can find some methods to address conditional modelling in mixed-integer programming.
- Solution Encoding of NSGA is very unclear.
- Three methods have been proposed but I can not see sufficient comparison.
Author Response
We are grateful for your review of our article. We implemented your comments and saw the improvement in the revised manuscript. Your comments are addressed point-by-point as follows,
- The title of the paper can be defined better. “Automatization of Carsharing” not only is a bit unclear but also, I think is off-topic. I had a look at some papers by the first author. I think this paper has been extracted from his thesis. Other papers by these authors on this topic have more relevant titles in comparison with the title of this paper. My suggestion is to revise it.
Reply: We understand your point, and revise the name to “Optimization” as reflected in the new title.
- The research problem is a bit unclear in the abstract. In addition, this is an optimization problem, the objective should be clarified in the abstract.
Reply: The abstract has been rewritten to emphasize the problem and its objectives.
- In line 34 “As a result, station-based services are less flexible, with an advantage of easier implementation and management.” This sentence can be considered as a good motivation for this study. Please elaborate it more and bring example(s) if it is possible.
Reply: Examples of relocation in bike-sharing and a charging station are given to elaborate the advantage of the station-based services.
- In line 37, “In this work, a new method to automatizethe fleet placement……” this word “automatize” has been repeated many times in the paper, but I believe it is not relevant to the paper at all. This study is an optimization problem and nothing else.
Reply: This is noted and these words have been revised.
- Figure 1 and its explanations in the body of the paper is unclear. I believe this figure can be presented in a better way.
Reply: We decide to remove this figure and focus on integrating this part in the research question instead.
- Section 3, particularly, Section 3.1 and 3.2 are very unclear. please see similar papers in this research area. The most common way is to bring a research problem (or statement) and explain the problem clearly (preferably using figures). Then bring one or several tables and list the parameters, set, decision variables, and then present the model. In addition, the proposed model has some problems, for example, types of decision variables have not been defined (continues, binary, ….) in addition the model C2 should be modelled in a proper way. You can find some methods to address conditional modelling in mixed-integer programming.
Reply: Previous Section 3 is now separated into two sections, Optimization Models and Optimization Methods to make it easier to read. The now Section 3, Optimization Model, is revised and figures are added to make it more comprehensive along with the newly formulated model. We also removed the correlation between objectives to make the section more compact and concise.
- Solution Encoding of NSGA is very unclear.
Reply: This paragraph is rewritten to describe the process better and in the spirit of the accompanied figure.
- Three methods have been proposed but I cannot see sufficient comparison.
Reply: Summary Tables for result comparison in instance LU2 and MU1 are added to emphasize the performance of each presented method.
Reviewer 2 Report
The article deals with the important problem of the location of carsharing stations. The presented theoretical model is clear and well explained
The article is too extensive and detailed in some parts.
In section 3.4.3 it is unnecessary to explain the meaning of the correlation coefficient. It is also not clear for what purpose the coefficient was determined, since the criteria are considered conflicting in advance.
The calculation procedures for the example instances are too detailed and it is difficult to focus on the key elements.
Are three instances necessary to achieve the purpose of the article? It seems that one representative example would be clearer and the article would become more compact.
If three instances have to be analyzed, the calculation results should be presented in collective statements and comparative tables.
Author Response
Thank you for your positive remarks and improvement suggestions. We have taken all of your comments into consideration and address them as follow,
- In section 3.4.3 it is unnecessary to explain the meaning of the correlation coefficient. It is also not clear for what purpose the coefficient was determined, since the criteria are considered conflicting in advance.
Reply: We removed this paragraph to make the section more compact and concise. Previous Section 3 is now separated into two sections to make the contents in the same section more relevant to each other. The now section 3 is revised with additional figures and improved optimization model definition.
- The calculation procedures for the example instances are too detailed and it is difficult to focus on the key elements.
Reply: Thank you very much for this positive remark and we see that the details could be useful to some readers who is theory-oriented. The details might be making it a bit more difficult to focus, but at the same time, have their worth. If it is not too negative in your opinion, we would like to keep them.
- Are three instances necessary to achieve the purpose of the article? It seems that one representative example would be clearer and the article would become more compact.
Reply: In our opinion, the three instances are crucial to establish the threshold where exact methods are not applicable anymore (which are instance LU1 and LU2). Then the third instance, MU1, is the showcase of real-world application and how the optimization methods fare in the real-world scenario which is not obvious in the instance LU2. Hence, we included all three instances.
- If three instances have to be analyzed, the calculation results should be presented in collective statements and comparative tables.
Reply: We added summary tables in addition to the pareto fronts figure for reader to easily compare each method as you suggested.
Round 2
Reviewer 1 Report
Still i think some concerns should be adressed:
I think the proposed crossover is weak And maybe it cannot guarantee to find desirable solutions because the crossover operator of this algorithm has been designed very bad. Just one small example, assume one parent just has “5” in its string and another one just “10”. Definitely, the children only have either 5 or 10. It means after several iterations, and many solutions will trap in a local optimum. It is true that mutation can handle a part of this issue, but the mutation rate is low and probably can not repair all of these solutions. In contrast, a high rate of mutation may decrease the performance and convergence rate of the algorithm.
In addition, the model still needs to be revised. For example, please use ∀ in your model.
Author Response
Dear Reviewer,
In response to your comment on the work, we acknowledge and are grateful for your comment. Please find the reply for your comments below.
I think the proposed crossover is weak And maybe it cannot guarantee to find desirable solutions because the crossover operator of this algorithm has been designed very bad. Just one small example, assume one parent just has “5” in its string and another one just “10”. Definitely, the children only have either 5 or 10. It means after several iterations, and many solutions will trap in a local optimum. It is true that mutation can handle a part of this issue, but the mutation rate is low and probably can not repair all of these solutions. In contrast, a high rate of mutation may decrease the performance and convergence rate of the algorithm.
Reply: We are also aware of this fact and believe that all evolutionary operators inherit this shortcoming. During this research, several combinations of crossover and mutations have been experimented and none resulted in guarantee of superiority over all others. This work focuses on implementation of FPP model rather than finding suitable parameters and operators for metaheuristic algorithm (i.e. NSGA-II) under consideration. Therefore, it may be possible that adjustment and tuning of evolutionary operators for their suitability may be necessary for its application in some instances which share too few characteristics with instances in this work.
We include the above clarification at the end of NSGA-II description in the revised manuscript. I sincerely trust that it is sufficient.
In addition, the model still needs to be revised. For example, please use ∀ in your model.
Reply: We addressed the lack of the "for all" signs in the model. We also add additional constraints to explicitly display the range of possible values for parameters and variables in the model to make the model more comprehensible.
Once again, your comments are well noted and appreciated.
Yours faithfully,
Round 3
Reviewer 1 Report
Accept.
Congratulation!
This manuscript is a resubmission of an earlier submission. The following is a list of the peer review reports and author responses from that submission.